# The Relation between Functional Performance, Falls and Previous Falls Among Participants in the Otago Programme: A Secondary Data Analysis

**DOI:** 10.3390/ijerph18126501

**Published:** 2021-06-16

**Authors:** María Consuelo Company-Sancho, Emma Alonso-Poncelas, Manuel Rich-Ruiz, María Ángeles Cidoncha-Moreno, Ana Gonzalez-Pisano, Eva Abad-Corpa

**Affiliations:** 1Health Promotion Service, Directorate General for Public Health, Canary Islands Health Service, 35003 Las Palmas, Spain; 2Quality Department, Lanzarote Health Services Management, Canary Islands Health Service, 35500 Arrecife, Spain; emmalopon@gmail.com; 3Maimonides Institute for Biomedical Research (IMIBIC), University of Cordoba (UCO), Hospital Universitario Reina Sofía (HURS), 14004 Cordoba, Spain; 4CIBER on Frailty and Healthy Ageing (CIBERFES), Instituto de Salud Carlos III, 28029 Madrid, Spain; eva.abad@um.es; 5General Head Office of Osakidetza, Basque Health Service, Subdirection of Nursing, IIS Bioaraba, 01006 Vitoria-Gasteiz, Spain; MARIAANGELES.CIDONCHAMORENO@osakidetza.eus; 6Health Service of the Principado of Asturias, 33011 Oviedo, Spain; anapisano@gmail.com; 7IMIB, Reina Sofía Hospital-University of Murcia, 30120 Murcia, Spain

**Keywords:** falls, elderly, community, primary care, prevention

## Abstract

Fall prevention is a key priority in healthcare policies. Multicomponent exercises reduce the risk of falls. The purpose of this study is to describe the relationship between functional performance and falls after following the Otago multicomponent exercise programme and previous falls. A prospective multi-centre intervention study was performed on 498 patients aged over 65 in primary care, with or without a history of previous falls. Sociodemographic, anthropometric and functionality data were collected. The primary outcome was the occurrence of falls; functional performance was measured using the Tinetti, Short Physical Performance Battery and Timed Up and Go tests. Among the patients, 29.7% referred to previous falls. There was a statistically significant (*p* < 0.001) increase in falls at 6 months (10.1%) and at 12 months (7.6%) among participants with previous falls in the baseline assessment compared to those without. In addition, the existence of previous falls could be considered a risk factor at 6 and 12 months (OR =2.37, *p* = 0.002, and OR = 1.76, *p* = 0.046, respectively). With regard to balance and gait, differences between the groups were observed at 6 months in the Tinetti score (*p* < 0.001) and in the baseline assessment Timed Up and Go score (*p* < 0.044). Multicomponent exercises improve the fall rate, balance and gait in older people, although this improvement is less in people with previous falls. Earlier intervention and tailoring of exercises in patients with previous falls could help improve outcomes.

## 1. Introduction

Demographic ageing is one of the main challenges facing Europe. It involves an increase in disability and dependence, in which frailty is a pre-disability state and a good predictor of adverse events [1]. The WHO describes this as “progressive age-related decline in physiological systems that results in decreased reserves of intrinsic capacity, which confers extreme vulnerability to stressors and increases the risk of a range of adverse health outcomes”. It can be prevented and treated to promote a long, healthier life [2].

According to reports by Joint Action “724099/ADVANTAGE”, the prevalence of frailty, reported in multiple studies on community cohort samples in participating countries, ranges from 2% to 60%, depending on factors such as the age of the study population and the frailty assessment instrument or classification used [2]. Fall prevention, directly associated with frailty, is one of the areas proposed as a priority for the health policy [1]. It plays a key role in preventing fractures among the elderly [3].

The WHO defines falls as an involuntary event in which the person loses balance and comes to rest on the ground or other firm surface. Generally, the cause is extrinsic and obvious and does not require thorough assessment [4]. Among the elderly population, the risk factors for falls include alterations to gait, loss of balance, strength and functional capacity, prior falls and fear of falling. Lower Activities of Daily Living (ADL) scores and older age are associated with more falls [5].

In Spain, 30% of over 65 will have a fall, with sequelae in 70% of cases, which will be severe in 10%. Women are at greater risk of falls and also suffer more severe consequences [1]. Falls also have a major financial impact due to the cost of additional treatment for injury (61%) and hospital admissions (12.3 extra days). In addition, 0.19 per 1000 occupied bed days are due to moderate or severe injury or death from falls among inpatients aged over 65 [6].

Recurrent falls are defined as two or more falls a year and differ from single falls in that they usually have intrinsic causes. Multiple falls are markers for other underlying factors or deficiencies. It is estimated that 50% of elderly people who have a fall will have another one the following year, with a greater likelihood of longer hospital stays and ending their days in a nursing home [4]. Recurrent falls also make elderly people more prone to disability, dependency and multiple diseases, while the incidence and, therefore, the consequences are reduced if preventative measures are applied [7]. Molinero concludes that having previous falls is an important risk marker for further falls (RR 1.67; 95% CI 1.13–2.46; *p* = 0.010) [8].

Falls in people aged over 65 with a history of falls range from 18.4% to 34.7% [4,9,10]. Furthermore, elderly people with recurrent falls have a greater percentage of falls involving injury (41% vs. 19%, *p* < 0.05) and greater subsequent disability in activities of daily living (ADL) in their physical, instrumental and social spheres [4]. 

These can trigger feelings of anxiety and fear of further falls, thereby producing “post-fall syndrome”, which involves the loss of self-confidence [11]. 

Multiple component interventions (a set combination of interventions such as education and education on falls) reduce the rate, number of people who fall and recurrent falls in the community, although no reduction in fractures or admissions was observed [12]. The multicomponent physical activity aims to improve endurance, flexibility, balance and muscle strength. Measuring functional performance through execution tests helps estimate the likelihood of falls.

The Otago Exercise Programme (OEP), developed by the University of Otago Medical School, is a multicomponent intervention designed to prevent falls, recommended for community-dwelling elderly people. It works on muscle strengthening, balance and endurance exercises. Several studies acknowledge it as an effective way to improve physical function and reduce falls and mortality [13,14]. Exercise-based interventions among community-dwelling elderly patients probably reduce the fear of falls to a limited extent immediately after the intervention, without increasing the risk or frequency of falls [15].

All of the interventions described above considerably reduce the incidence and, therefore, consequences of falls (RR 0.68; 95% CI 0.56–0.79; *p* < 0.001) [14]. It is thus worthwhile establishing the preventative capacity of such programmes among elderly patients with specific characteristics, such as recurrent falls. Therefore, our objective is to describe the relationship between functional performance and falls after following the Otago multicomponent exercise programme and previous falls.

## 2. Materials and Methods

This study was a secondary analysis of data collected in a non-inferiority multi-centre intervention controlled clinical trial comparing two study groups (group vs. individual training sessions) in a community-dwelling 65- to 80-year-old population. The study was recorded in ClinicalTrials.org (NCT03320668). The OEP was applied between September 2017 and December 2019. The sample size was calculated assuming one-tailed testing, with an alpha error of 0.025 and 1-beta of 0.80, assuming a minimum decrease of 15% in falls, a non-inferiority margin of 10% and assuming 10% lost in follow-up, leaving a total of 728 individuals. Inclusion criteria: subjects are registered with their health area, aged between 65 and 80 years old, not institutionalised, has a Berg balance score of ≥ 33 and signed informed consent. Exclusion criteria: people who had been living outside the area covered by the primary health centre for more than 9 months, people with moderate to severe cognitive impairment, vision or hearing impairment, or absolute contraindication for exercise.

The study participants were incorporated in an OEP, which consisted of five sessions in weeks 1, 2, 4 and 8 and a reinforcement session after 6 months. The intervention was considered incomplete (and, therefore, lost to follow-up) when the subject did not complete 100% of the five sessions. The participants performed the exercises for 30 min twice a week or incorporated them into their daily routine. They received information on the exercises and the materials required to perform them (weights). They were provided with reminders using a text-messaging alert system and phone calls following a predetermined protocol. Subjects who did not complete the intervention were considered lost in follow-up. The OEP was provided by primary care professionals (mostly nurses) through cascade training proposed by Later Life Training (Otago-LLT). It was applied to patients in 8 autonomous communities and 10 centres.

The hypothesis tested in this secondary analysis was that the number of falls is higher and functional performance lower in patients aged 65 years and older with previous falls after following the Otago multicomponent exercise programme.

The primary outcomes were the occurrence of falls and performance on the Tinetti gait and balance, Short Physical Performance Battery (SPBB) and Timed Up and Go (TUG) tests. The primary independent variable in this secondary analysis was a history of falls in the previous 12 months. Baseline data were also collected on sociodemographic variables (age, sex, education level, marital status, type of domestic arrangement), anthropometric variables (weight, size, body-mass index (BMI)) and functionality, using the Barthel Index of Activities of Daily Living (ADL) and the Lawton scale. 

Three measurements were taken: one at the baseline and then at 6 and 12 months (with regard to the previous 6 months) from the start of the intervention, ascertaining the number of falls and the Tinetti, SPBB and TUG test scores. An ad hoc questionnaire was designed for the data collection. 

The mean, standard deviation, median and 25th and 75th percentiles were calculated for the quantitative variables. The Kolmogorov-Smirnov test was used to determine the normality of the data distribution. Frequency and percentage were calculated for the qualitative variables. A Mann-Whitney U-test was used to assess the differences between numerical variables scores in groups of older people with and without previous falls. Fisher’s exact test was used to compare both groups when the variables were categorical. McNemar’s test was used for testing the significance of changes in dichotomous variables at two points in time. A logistic regression was used to predict dichotomous variables. Crude logistic regressions of the outcome variable with each risk factor were performed before running the final covariate-adjusted regression model. The statistics programme used was “R”, version 4.0.2 (Foundation for Statistical Computing, Vienna, Austria).

This study is part of the PI16CIII/00031 intramural project “Efficacy of the Otago Exercise Programme” to reduce falls in community-dwelling adults aged 65–80 years old when delivered as a group or individual training [16]. It was approved by the clinical research ethics committees at the institutions of each of the participating centres and was conducted according to the recommendations of the Declaration of Helsinki.

## 3. Results

### 3.1. Descriptive Statistics

The study sample included 498 patients (329 participants out of the 827 who met the inclusion criteria and agreed to participate were lost to follow-up), of whom 67.0% were women. The mean age was 71.8 (SD 4.15) years; 8.6% had no formal education, 21.3% had not completed primary education, 45.4% had completed primary education, 16.5% had secondary education and 8.2% had a university education; 63.9% were married and 25.7% were widowed. Of these, 65.1% lived with their partner and 24.5% lived alone. The mean weight of the participants was 74.9 kg (SD 12.92), the mean height was 159.78 cm (SD 8.75) and the mean BMI was 29.36 (SD 4.49).

With regard to the main independent variable, 29.7% (*n* = 148) of participants reported falls in the 12 months prior to the start of the study. Regarding functionality, 75% of the total sample had maximum Barthel, Lawton and Brody scores. 

With respect to falls during the OEP periods, 11.8% (*n* = 59) of participants had one or more falls after 6 months, this figure rising to 12.2% (*n* = 61) in the following 6 months. Regarding functional performance measures, the mean Tinetti scores at the baseline and 6 and 12 months were 31.8 (SD 4.08), 33.9 (SD 2.94) and 33.3 (SD 2.92), respectively. Only the mean scores of this test showed statistically significant differences when two-by-two comparisons were made (*p* < 0.001, in the case of baseline versus 6 and 12 months, and *p* = 0.042, in the case of 6 versus 12 months).

### 3.2. Bivariate Analysis 

With regard to users with or without falls in the baseline assessment, there were significant differences in independence for ADL (Barthel) at the baseline (*p* = 0.043) and at 6 months (*p* = 0.01), which was lower for elderly people with prior falls.

With respect to improvement in balance and gait as measured in the Tinetti, SPPB and TUG tests, differences between users with and without baseline falls were only found at 6 months in the Tinetti score, where the mean score was 33.37 (SD 2.68) in the “no previous falls” group and 32.41 (SD 3.39) in the “ previous falls” group with a *p*-value < 0.001, although at baseline there was already a mean of 32.19 (SD 3.91) in the “no previous falls” group and 31.16 (SD 4.40) in the “ previous falls” group with a *p*-value < 0.002

With regard to the TUG variable, significant differences were also observed in the baseline assessment (*p* = 0.044), with higher scores among elderly participants with previous falls, but not at 6 and 12 months. The SPPB test showed no significant differences in any of the measurements.

The elderly participants with previous falls had a higher frequency of falls during the study, both at 6 and 12 months, with percentages of 18.9% and 17.6%, respectively, compared to percentages of 8.9% and 10% in the group of elderly without previous falls (Table 1 and Table 2). The differences in both times periods are statistically significant (*p* < 0.001 and *p* < 0.001, respectively)

### 3.3. Multivariate Analysis

With regard to the logistic regression analysis of fall-associated factors, the existence of previous falls was significantly related to falls in both models. At 6 months, having previous falls is considered a risk factor (Odds Ratio (OR) =2.37; *p* = 0.002; Table 3). At 12 months, having a previous fall is marginally associated with the occurrence of falls (OR = 1.76; *p* = 0.046; Table 4). Living alone at baseline is considered a risk factor for a larger number of falls at the end of the study (*p* = 0.004), and age is marginally significant at 6 months (*p* = 0.060).

## 4. Discussion

The data describe a population with a high capacity for ADLs (according to the Barthel index and the Lawton scale). However, the group with previous falls has a significantly lower Barthel score at the baseline, which is not surprising considering the relationship between ADL dependence and falls [17] and could explain the higher dependence among those with previous falls.

Concerning balance and gait, the data also indicate baseline differences favouring those without previous falls in the Tinetti and TUG tests, which persist at six months for the Tinetti test. These results are similar to those from other studies, where older adults with previous falls have more significant functional limitations and more unsatisfactory results in performance-oriented mobility assessment (POMA) scores (such as Tinetti [4]). Therefore, the exercises should be adapted to the baseline situation (functional status), as suggested by Maren (2019) [18], and further adjusting them to changes in health and physical ability over time [19].

With regard to the baseline incidence of falls, our data match those from previous studies that report incidences around 25–35% before the start of the study. Concerning the decrease in falls at 12 months of intervention, our results (14%) are similar to those reported by other authors, such as Gawler [20]. However, this reduction in falls is not homogeneous between the groups with and without previous falls. Furthermore, our study identifies previous falls as a factor associated with falls at six months and 12 months.

In this sense, elderly care programmes and protocols should propose earlier action to provide fall prevention exercises to patients with a history of falls. Falls are more frequent in people with a history of falls, so prevention should be increased, including exercise programmes with proven efficacy in reducing falls.

Another factor associated with falls in our study results (at 12 months) is “living alone.” As also found in Rodriguez-Molinero [7], people living alone fall more often; hence, an accompanied living arrangement could be a community protection factor.

On the other hand, and even though the results of our study do not point to age as a factor associated with falls, as was reported in previous studies [21], exercise programmes and other activities of this type for the elderly should contemplate this risk factor as a critical element in preventing falls.

In this regard, our study found no association between falls and BMI. These data differ from previous studies showing an association between obesity and falls [22]. In Mitchell (2014), obese individuals had a 31% higher risk of falls, but not of fall-related injuries [23].

The approach to falls, along with aging and frailty, is a priority for research in Europe. The Advantage joint action, co-funded through the Third European Union Health Programme 2014–2020, is the first to attempt to prevent frailty [24]. It aims to define a common strategy for Europe to foster a more homogeneous approach to the issue.

In 2017, it released the “State of the art report on the prevention and management of frailty”, proposing multicomponent exercise programmes (such as the OEP) to improve functional capacity and health in frail people. The report points out that resistance, flexibility and balance training effectively reduce frailty [25,26]. The principal risk factor for frailty is inactivity, where physical activity-based interventions have proved effective in reversing frailty and disability, improving the cognitive status and fostering emotional well-being [27].

In line with WHO proposals, particular emphasis should be placed on including fall prevention actions in the public health service portfolio.

Primary care is the most appropriate level for detecting and managing frailty [28,29] and promoting fall prevention, especially among people with recurrent falls.

Regarding the study’s limitations, the fact that the face-to-face intervention ended at four months may have led to a loss of adherence, despite subsequent monthly telephone calls and face-to-face contact at 6 and 12 months. In this context, O’Shea links a long-term decrease in adherence to the removal of support groups and supervision [30]. This may have been the case in our study, as the face-to-face programme was short, lasting 4–6 months, after which there was no direct contact.

The short follow-up of the study subjects, lasting 12 months, may also be a limitation. If the different starting situations had been taken into account, the exercises could also possibly have been better adapted to participants with lower Tinetti and TUG scores, thereby improving the results in patients with previous falls.

Finally, although we did not reach our target sample size due to losses, this study included a large sample from an intervention trial in which protocol and intention to treat analyses were conducted in the principal analysis.

In any case, we believe that the results allow us to emphasise two characteristics that exercise programmes designed to prevent falls in elderly persons living at home should contain: the inclusion of therapeutic strategies adapted to the specific needs of the person (functional capacity or previous falls), and the maintenance of support groups and supervision that help to reduce lack of adherence to the programme [31].

## 5. Conclusions

Multicomponent exercises improve falls, balance and gait in patients with or without previous falls; however, a history of previous falls weakens improvement. Having previous falls is considered a risk factor at 6 and 12 months. Concerning balance and gait, the data also indicate baseline differences favouring those without previous falls in the Tinetti and TUG tests, which persist at 6 months for the Tinetti test. This result suggests that measures such as earlier intervention, adapting the OEP and tailoring exercises to the individual are recommendable and should be increased with age, taking into account the history of falls. Promoting both community and primary care actions, in accordance with European Union proposals, is advisable given the impact of falls on elderly people. 

## Figures and Tables

**Table 1 ijerph-18-06501-t001:** Difference in falls at 6 months in relation to the existence of previous falls *.

Previous Falls	Falls at 6 Months		
	No Falls	Falls	Row Total
No falls	319 91.1%	31 8.7%	350 70.3%
Falls	120 81.1%	28 18.9%	148 29.7%
Column Total	439	59	498

* Fisher’s exact test.

**Table 2 ijerph-18-06501-t002:** Difference in falls at 12 months in relation to the existence of previous falls *.

Previous Falls	Falls at 12 Months		
	No Falls	Falls	Row Total
No falls	315 90.0%	35 10.0%	350 70.3%
Falls	122 82.4%	26 17.6%	148 29.7%
Column Total	437	61	498

* Fisher’s exact test.

**Table 3 ijerph-18-06501-t003:** Crude and adjusted logistic regression models of falls associated factors at 6 months.

Variable	Crude Logistic Regression			Covariate-Adjusted Regression Model		
	OR	95% CI of OR	*p*	OR	95% CI of OR	*p*
Intercept	-	-	-	0.00	0.00–0.05	0.003
Baseline falls: Fall	2.40	1.38–4.18	0.002	2.37	1.35–4.14	0.002
Age	1.05	0.99–1.13	0.116	1.07	1.00–1.14	0.060
BMI	1.05	0.99–1.11	0.138	1.05	0.99–1.12	0.120
Education level: No formal education	1 (ref)			1 (ref)		
Incomplete primary education	0.86	0.31–2.61	0.780	--	--	--
Completed primary education	0.73	0.30–2.09	0.526	--	--	--
Secondary education	0.67	0.22–2.16	0.482	--	--	--
University education	1.49	0.47–4.97	0.496	--	--	--

**Table 4 ijerph-18-06501-t004:** Crude and adjusted logistic regression models of falls associated factors at 12 months.

Variable	Crude Logistic Regression			Covariate-Adjusted Regression Model		
	OR	95% CI of OR	*p*	OR	95% CI of OR	*p*
Intercept	-	-	-	0.09	0.06–0.13	0.001
Baseline falls: Fall	1.92	1.10–3.31	0.020	1.76	1.00–3.07	0.046
Age	1.00	0.94–1.07	0.982	--	--	--
BMI	1.02	0.96–1.08	0.554	--	--	--
Living arr.: alone	2.43	1.38–4.24	0.002	2.29	1.29–4.00	0.004
Education level: No formal education	1 (ref)			1 (ref)		
Incomplete primary education	0.85	0.33–2.38	0.740	--	--	--
Completed primary education	0.58	0.24–1.56	0.248	--	--	--
Secondary education	0.88	0.33–2.55	0.808	--	--	--
University education	0.56	0.14–2.00	0.380	--	--	--

## Data Availability

Data are available upon reasonable request.

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
