# Peer review of "The Relation between Functional Performance, Falls and Previous Falls Among Participants in the Otago Programme: A Secondary Data Analysis"

_ijerph, 2021, doi:10.3390/ijerph18126501_

Round 1

Reviewer 1 Report

This paper evaluated functional performance and falls amongst participants in the Otago exercise program. The authors found differences between those that have a history of falling and those that do not. Ultimately, while the language was grammatically fine, for the most part, I had a great deal of difficulty understanding the results, specifically which tests were used for which results. And, because of these challenges, it was difficulty evaluating how the results were interpreted. My biggest concern is that the methods (particularly the statistics section) be elaborated on, and results be clarified and elaborated on, so that the reader can better evaluate the manuscript. Below I list some more specific details as to the areas I found problematic that the authors should consider revising.

Line 25: Insert ‘between’ between ‘relation’ and ‘functional performance’

Line 36: This line is unclear in its meaning consider revising: ‘Exercise improves the rate of falls, balance and gait in people with falls, although previous falls lessen the improvement.’

Line 50: Consider inserting ‘the’ between ‘as’ and ‘age’

Line 64: Change ‘usual’ to ‘usually’

Line 96: Consider removing the first ‘data’ in the sentence

Line 103: It is unclear what is meant by ‘health area’. Is this a regional registration?

Line 128: Consider rewording…maybe “Test was used to determine the normality of the data…or test was used to determine whether the date were normally distributed”

Line 130: It is unclear here what the Mann-Whitney U-test was used for? It is unclear what “check the numerical variables means”

Line 131: Fishers exact test was used to test for statistical significant between qualitative variables? Maybe clarify or reword

Line 132: Consider rewording this line to be more specific. Was McNemar’s test used to test for marginal homogeneity? The term ‘check’ is not very informative consider rewording.

Line 134: Isn’t the programming language simply R, and R Core Team the citation for the software language? Consider clarifying this in the text.

Line 143: What was the education of the remaining 33.23% ?

Line 145: The mean baseline and end weight was identical in mean and standard deviation between the two measure taken at different times? Is the end measure 12 months later? The two values being identical suggests seems surprising stability over time…

Line 149 – 152: Are these changes statistically significant or observed trends? Consider clarifying this in the text.

Line 150-151: The authors should comment here on the size of the decrease at 12 mo (99.47 vs 99.42) and whether this a result of natural variability or a true decrease and if this is a true decrease if it is large enough to have clinical/real world relevance.

Line 153-154: Does referred to falls mean these participants reported falls in the 12 months prior? If so consider rewording.

Line 156: Is ‘after 6 months’ referenced to the start of the training program?

Line 157-160: Are these observed trends or statistical differences? Maybe clarify with minor rewording of the text. Seems like these are simple descriptive reports and not inferences drawn from statistical tests.

Line 164-166: Does this mean that there were differences in independence scores between those with a history of falls and those that did not? Also, is the extra ‘in’ in ‘with in independence’ supposed to be there?

Line 169-170: Was the baseline effect considered in the comparison at 6 months? Is the difference at 6 months still significant if baseline is considered

Line 175 (Figure 1): What are the three groups. Please include a more detailed description in the caption

Line 175 Figure 1: Data appear highly skewed, clarify which statistical tests were used to compared these, and how was the data distribution was considered… This could maybe be part of the methods where the statistical tests are summarized. Also, please indicate on the figure which differences are statistically different and consider adding which statistical tests were used in the caption.

Line 177: A statistical difference in the number of falls? Please clarify these lines in the text so the reader doesn’t have to refer to the table to know what the comparison is.

Lines 181-187: Text alignment seems off…Maybe a formatting issue with the publisher?

Line 183: The existence of previous fall remained a significant predictor for both the 6 and 12 month models? Also, what was alpha for these models? It seems from the statement that p=0.052 is significant? Is p > 0.05 (maybe I missed this)? Line 185 says p = 0.07 is close to significance which leads me to suspect that p = 0.05 in which case falls were not a significant predictor in the 12 month model (language could be more precise here, though I recognize the arbitrariness of a cutoff such as this and the impact it has on what is significant and what is not)?

Table 3 and 4 seems to have the text boxes filled with white creating white rectangles behind the text. This was a bit distracting and maybe unnecessary. Consider removing the background fill. Also, in the p column the age p-value is blue. Is there a reason for this (similar for the p-value for living alone in table 4)? Maybe this is a formatting problem with the publisher? Consider adding a table caption to clarify what is being highlighted in the table.

Table 3 and 4: It is unclear, to me at least, why each table presents univariate and multivariate results and from which tests these results came from. Please clarify this in the methods, results and/or in the table captions.

Line 184: This seems to be the only mention of correlation, please clarify this statement. Again it is unclear what analyses are being performed. Is this from the logistic regression… are the correlation coefficients presented? Were both analyses performed?

Line 192: The adaptive daily living scores improved at 6 months? Maybe clarify which scores specifically.

Line 194: Please elaborate on ‘the data reveal falls’

Line 199: maybe the group with a history of falls?

Line 229: Multiple periods?

Line 230-233: Why might the authors have observed no difference? Also, the however at the start implies a contrast to something said previously but the contrast follows this statement, consider revising.

Line 249: Opportunity for what?

Author Response

Consulte el archivo adjunto.

Reviewer 2 Report

Thank you for inviting me to review this manuscript. The topic of the present study is of interest. Indeed, preventing falls in order to reduce their health and economic consequences is needed. However, in the present form, the paper presents several limitations. Proper literature on the topic is not presented, the objective is not clear, the statistical design and the results should be better presented, and the discussion is short and hard to follow.

 objective is unclear

Abstract :

  • the third sentence, describing the objective may seem incomplete/unclear

Introduction :

  • Most references are not in the English language and do not seem to be the most appropriate.
  • L69-70 – Please revise the last sentence.
  • The objective and its novelty are not clear.

Methods :

  • L 101- Sample size calculation: the reason why a non-inferiority margin was used is not clear, as no non-inferiority hypothesis was presented before.
  • L 130 – In the statistics section, I believe the tested hypothesis should be briefly presented.

Results

  • L- 142 - The reason why 728 participants were needed (sample size calculation), but only 498 were included should be explained.
  • The statistical design is not clear and should be better explained
  • Statistical results are not presented properly, and the name of the test that resulted in the presented results should be mentioned, only presenting the p-value does not seem enough.
  • Tables (in particular Table 3 and 4) are poorly presented.

Discussion

  • Because the objective is not clear, it is difficult to determine if the discussion properly discusses the confirmation of the hypothesis.
  • The discussion should be better structured.

Round 2

Reviewer 1 Report

Line 143: Include here what is meant by the crude and adjusted models? Are the authors referring to a model excluding and including an intercept?

Line 198: Does ‘OR’ refer to odds ratio? Is so consider defining this at its first use.

Line 218: ‘Exercices’ is misspelled
